# HETEROGENEOUS BITWIDTH BINARIZATION IN CONVOLUTIONAL NEURAL NETWORKS

## ABSTRACT

Recent work has shown that performing inference with fast, very-low-bitwidth (e.g., 1 to 2 bits) representations of values in models can yield surprisingly accurate results. However, although 2-bit approximated networks have been shown to be quite accurate, 1 bit approximations, which are twice as fast, have restrictively low accuracy. We propose a method to train models whose weights are a mixture of bitwidths, that allows us to more finely tune the accuracy/speed trade-off. We present the "middle-out" criterion for determining the bitwidth for each value, and show how to integrate it into training models with a desired mixture of bitwidths. We evaluate several architectures and binarization techniques on the ImageNet dataset. We show that our heterogeneous bitwidth approximation achieves super-linear scaling of accuracy with bitwidth. Using an average of only 1.4 bits, we are able to outperform state-of-the-art 2-bit architectures.

## 1 INTRODUCTION

With Convolutional Neural Nets (CNNs) now outperforming humans in vision classification tasks (Szegedy et al., 2015), it is clear that CNNs will be a mainstay of AI applications. However, CNNs are known to be computationally demanding, and are most comfortably run on GPUs. For execution in mobile and embedded settings, or when a given CNN is evaluated many times, using a GPU may be too costly. The search for inexpensive variants of CNNs has yielded techniques such as hashing (Chen et al., 2015), vector quantization (Gong et al., 2014), and pruning (Han et al., 2015). One particularly promising track is binarization (Courbariaux et al., 2015), which replaces 32-bit floating point values with single bits, either +1 or −1, and (optionally) replaces floating point multiplies with packed bitwise popcount-xnors Hubara et al. (2016). Binarization can reduce the size of models by up to $32\times$, and reduce the number of operations executed by up to $64\times$.

Binarized CNNs are faster and smaller, but also less accurate. Much research has therefore focused on reducing the accuracy gap between binary models and their floating point counterparts. The typical approach is to add bits to the activations and weights of a network, giving a better approximation of the true values. However, the cost of extra bits is quite high. Using $n$ bits to approximate just the weights increases the computation and memory required by a factor of $n$ compared to 1-bit binarization. Further using $n$ bits to approximate activations as well requires $n^2$ times the resources as one bit. There is thus a strong motivation to use as few bits as possible while still achieving acceptable accuracy. However, today's binary approximations are locked to use the same number of bits for all approximated values, and the gap in accuracy between bits can be substantial. For example, recent work concludes 1-bit accuracy is unsatisfactory while 2-bit accuracy is quite high (Tang et al., 2017) (also see Table 1).

In order to bridge the gap between integer bits, we introduce Heterogeneous Bitwidth Neural Networks (HBNNs), which use a mix of integer bitwidths to allow values to have effectively (i.e., on average) fractional bitwidths. The freedom to select from multiple bitwidths allows HBNNs to approximate each value better than fixed-bitwidth schemes, giving them disproportionate accuracy gains for the number of effective bits used. For instance, Alexnet trained with an average of 1.4 bits has comparable (actually, slightly higher) accuracy to training with a fixed two bits (Table 1).

Our main contributions are:

(1) We propose HBNNs as a way to break the integer-bitwidth barrier in binarized networks.

(2) We study several techniques for distributing the bitwidths in a HBNN, and introduce the *middle-out bitwidth selection algorithm*, which uses the full representational power of heterogeneous bitwidths to learn good bitwidth distributions.

(3) We perform a comprehensive study of heterogeneous binarization on the ImageNet dataset using an AlexNet architecture. We evaluate many fractional bitwidths and compare to state of the art results. HBNNs typically yield the smallest and fastest networks at each accuracy. Further, we show that it is usually possible to equal, or improve upon, 2-bit-binarized networks with an average of 1.4 bits.

(4) We show that heterogeneous binarization is applicable to MobileNet Howard et al. (2017), demonstrating that its benefits apply even to modern, optimized architectures.

## 2 HOMOGENEOUS NETWORK BINARIZATION

In this section we discuss existing techniques for binarization. Table 1 summarizes their accuracy.[1]

When training a binary network, all techniques including ours maintain weights in floating point format. During forward propagation, the weights (and activations, if both weights and activations are to be binarized) are passed through a *binarization function* $\mathcal{B}$, which projects incoming values to a small, discrete set. In backwards propagation, a *custom gradient*,which updates the floating point weights, is applied for the binarization layer,. After training is complete, the binarization function is applied one last time to the floating point weights to create a true binary (or more generally, small, discrete) set of weights, which is used for inference from then on.

Binarization was first introduced by Courbariaux et al. (2015). In this initial investigation, dubbed BinaryConnect, 32-bit tensors $T$ were converted to 1-bit variants $T^B$ using the stochastic equation

$$\mathcal{B}(T) \triangleq T^B = \begin{cases} +1 & \text{with probability } p = \sigma(T), \\ -1 & \text{with probability } 1 - p \end{cases} \tag{1}$$

where $\sigma$ is the hard sigmoid function defined by $\sigma(x) = \max(0, \min(1, \frac{x+1}{2}))$. For the custom gradient function, BinaryConnect simply used $\frac{dT^B}{dT} = 1$.

Although BinaryConnect showed excellent results on relatively simple datasets such as CIFAR-10 and MNIST, it performed poorly on ImageNet, achieving only an accuracy of 27.9%. Courbariaux et al. (2016) later improved this model by simplifying the binarization by simply taking $T^B = \text{sign}(T)$ and adding a gradient for this operation, namely the *straight-through estimator*:

$$\frac{dT^B}{dT} = 1_{|T| \leq 1}. \tag{2}$$

The authors showed that the straight-through estimator further improved accuracy on small datasets. However, they did not attempt to train a model ImageNet in this work.

Rastegari et al. (2016) made a slight modification to the simple pure single bit representation that showed improved results. Now taking a binarized approximation as

$$T^B = \alpha_i \text{sign}(T) \quad \text{with} \quad \alpha_i = \frac{1}{d} \sum_{j=1}^{d} |T_j|. \tag{3}$$

This additional scalar term allows binarized values to better fit the distribution of the incoming floating-point values, giving a higher fidelity approximation for very little extra computation. The addition of scalars and the straight-through estimator gradient allowed the authors to achieve an accuracy on ImageNet, 44.2% Top-1, a significant improvement over previous work.

Hubara et al. (2016) and Zhou et al. (2016) found that increasing the number of bits used to quantize the activations of the network gave a considerable boost to the accuracy, achieving similar Top-1 accuracy of 51.03% and 50.7% respectively. The precise binarization function varied, but the typical approaches include linearly placing the quantization points between 0 and 1, clamping values below a threshold distance from zero to zero (Li et al., 2016), and computing higher bits by measuring

---

[1]In line with prior work, we use the AlexNet model trained on the ImageNet dataset as the baseline.

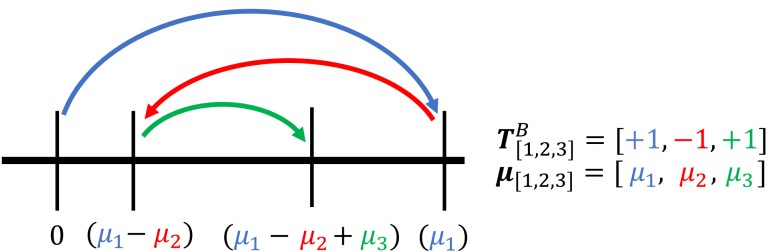

Figure 1: Residual error binarization with $n = 3$ bits. Computing each bit takes a step from the position of the previous bit (see Equation 4).

the residual error from lower bits (Tang et al., 2017). All $n$-bit binarization schemes require similar amounts of computation at inference time, and have similar accuracy (see Table 1). In this work, we extend the *residual error binarization* function Tang et al. (2017) for binarizing to multiple $(n)$ bits:

$$T_1^B = \text{sign}(T), \ \mu_1 = \text{mean}(|T|)$$

$$E_n = T - \sum_{i=1}^{n} \mu_i \times T_i^B$$

$$T_{n>1}^B = \text{sign}(E_{n-1}), \ \mu_{n>1} = \text{mean}(|E_{n-1}|) \quad (4)$$

$$T \approx \sum_{i=1}^{n} \mu_i \times T_i^B$$

where $T$ is the input tensor, $E_n$ is the residual error up to bit $n$, $T_n^B$ is a tensor representing the $n^\text{th}$ bit of the approximation, and $\mu_n$ is a scaling factor for the $n^\text{th}$ bit. Note that the calculation of bit $n$ is a recursive operation that relies on the values of all bits less than $n$. Residual error binarization has each additional bit take a step from the value of the previous bit. Figure 1 illustrates the process of binarizing a single value to 3 bits. Since every binarized value is derived by taking $n$ steps, where each step goes left or right, residual error binarization approximates inputs using one of $2^n$ values.

To date, there remains a considerable gap between the performance of 1-bit and 2-bit networks (compare rows 7 and 9 of Table 1). The highest full (i.e., where both weights and activations are quantized) single-bit performer on AlexNet, Xnor-Net, remains roughly 7 percentage points less accurate (top 1) than the 2-bit variant, which is itself about 5.5 points less accurate than the 32-bit variant (row 16). When only weights are binarized, very recent results (Dong et al., 2017) similarly find that binarizing to 2 bits can yield nearly full accuracy (row 2), while the 1-bit equivalent lags by 4 points (row 1). The flip side to using 2 bits for binarization is that the resulting models require double the number of operations as the 1-bit variants at inference time.

These observations naturally lead to the question, explored below, of whether it is possible to attain accuracies closer to those of 2-bit models while running at speeds closer to those of 1-bit variants. Of course, it is also fundamentally interesting to understand whether it is possible to match the accuracy of higher bitwidth models with those that have lower (on average) bitwidth.

## 3 HETEROGENEOUS BINARIZATION

In this section, we discuss how to extend residual error binarization to allow heterogeneous (effectively fractional) bitwidths. We develop several different methods for distributing the bits of a heterogeneous approximation. We point out the inherent representational benefits of heterogeneous binarization. Finally, we discuss how HBNNs could be implemented efficiently to benefit from increased speed and compression.

### 3.1 HETEROGENEOUS RESIDUAL ERROR BINARIZATION

We modify Equation 4 , which binarizes to $n$ bits, to instead binarize to a mixture of bitwidths by changing the third line as follows:

$$T_{n>1}^B = \text{sign}(E_{n-1,j}), \ \mu_{n>1} = \text{mean}(|E_{n-1,j}|), \ \text{with } j : M_j \geq n \quad (5)$$

Note that the only addition is tensor $M$, which is the same shape as $T$, and specifies the number of bits $M_j$ that the $j^{\text{th}}$ entry of $T$ should be binarized to. In each round $n$ of the binarization recurrence, we now only consider values that are not finished binarizing, i.e, which have $M_j \geq n$. Unlike homogeneous binarization, therefore, heterogeneous binarization generates binarized values by taking *up to*, not necessarily exactly, $n$ steps. Thus, the number of distinct values representable is $\sum_{i=1}^{n} 2^n = 2^{n+1} - 2$, which is roughly double that of the homogeneous case.

In the homogeneous case, on average, each step improves the accuracy of the approximation, but there may be certain individual values that would benefit from not taking a step, in Figure 1 for example, it is possible that $(\mu_1 - \mu_2)$ approximates the target value better than $(\mu_1 - \mu_2 + \mu_3)$. If values that benefit from not taking a step can be targeted and assigned fewer bits, the overall approximation accuracy will improve despite there being a lower average bitwidth.

## 3.2 BIT SELECTION METHODS

The question of how to distribute bits in a heterogeneous binary tensor to achieve high representational power is equivalent to asking how $M$ should be generated. When computing $M$, our goal is to take a set of constraints indicating what fraction of $T$ should be binarized to each bitwidth, perhaps 70% to 1 bit and 30% to 2 bits for example, and choose those values which benefit most (or are hurt least) by not taking additional steps. Algorithm 1 shows how we compute $M$.

---

**Algorithm 1** Generation of bit map $M$.

**Input:** A tensor $T$ of size $N$ and a list $P$ of tuples containing a bitwidth and the percentage of $T$ that should be binarized to that bitwidth. $P$ is sorted by bitwidth, smallest first.
**Output:** A bit map $M$ that can be used in Equation 5 to heterogeneously binarize $T$.

1:   $R = T$          $\triangleright$ Initialize $R$, which contains values that have not yet been assigned a bitwidth
2:   $x = 0$
3:   **for** $(b, p_b)$ in $P$ **do**       $\triangleright$ $b$ is a bitwidth and $p_b$ is the percentage of $T$ to binarize to width $b$.
4:      $S = \text{select}(R)$       $\triangleright$ Sort indices of remaining values by suitability for $b$-bit binarization.
5:      $M[S[x : x + p_b N]] = b$
6:      $R = R \setminus R[S[x : x + p_b N]]$          $\triangleright$ Do not consider these indices in next step.
7:      $x \mathrel{+}= p_b N$
8: **end for**

---

The algorithm simply steps through each bitwidth $b$ (line 3), and for the corresponding fraction $p_b$ of values to be binarized to $b$ bits, selects (lines 4 and 5) the "most suitable" $p_b N$ values of $T$ to be binarized to $b$ bits. Once values are binarized, they are not considered in future steps (line 6). We propose several simple methods as candidates for the select function: Top-Down (TD), Middle-Out (MO), Bottom-Up (BU), and Random (R) selection. The first three techniques depend on the input data. They pick the largest, closest-to-mean or smallest values. The last technique is oblivious to incoming data, and assigns a fixed uniform pattern of bitwidths.

$$
\begin{aligned}
\text{TD}(T) &= \text{sort}(|T|, \text{descending}) \\
\text{MO}(T) &= \text{sort}(|T| - \text{mean}(|T|), \text{ascending}) \\
\text{BU}(T) &= \text{sort}(|T|, \text{ascending}) \\
\text{R}(T) &= \text{a fixed uniformly random permutation of } T
\end{aligned}
\tag{6}
$$

The intuition for Middle-Out derives from Figure 1, where we see that when a step is taken from the previous bit, that previous bit falls in the middle of the two new values. Thus, the entries of $T$ that most benefit from not taking a step are those that are close to the center (or "middle") of the remaining data. This suggests that fixing values near the middle of the distribution to a low bitwidth will yield the best results. Our results show that MO is much better than the other techniques.

## 3.3 IMPLEMENTABILITY

The typical appeal of binary networks is that they reduce model size and the number of computations needed. Model size is reduced by replacing 32-bit-float weights with a small number of bits and packing those bits into 64-bit integers. Computation reduction becomes possible when both inputs

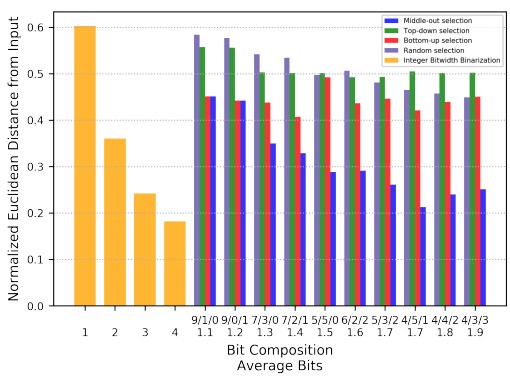
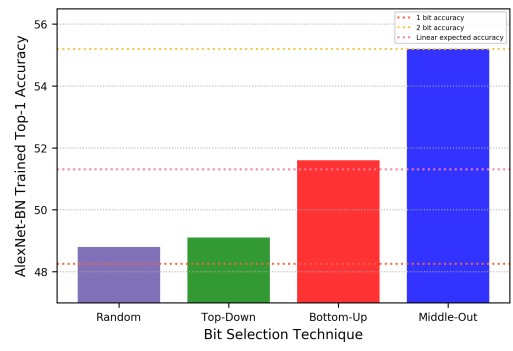

(a) Bit selection representational power.    (b) 1.4 bit HBNN AlexNet Accuracy.

Figure 2: Effectiveness of heterogeneous bit selection techniques (a) ability of different binarization schemes to approximate a large tensor of normally distributed random values. A bit composition denoted as x/y/z indicates x% of values are binarized to 1-bit, y% to 2-bit, and z% to 3-bit. (b) accuracy of 1.4 bit heterogeneous binarized AlexNet-BN trained using each bit-selection technique.

and weights are binarized Hubara et al. (2016). This allows floating point multiplications to be replaced by popcount-xnor operations (which is the bit equivalent of a multiply accumulate). A single popcount-xnor on packed 64-bit inputs does the work of 64 multiply accumulates. However, because heterogeneous bitwidth tensors are essentially sparse, they can not be efficiently packed into integers. Both packing and performing xnor-popcounts on a heterogeneous tensor would require an additional tensor like $M$ that indicates the bitwidth of each value. However, packing is only needed because CPUs and GPUs are designed to operate on groups of bits simultaneously. In custom hardware such as an ASIC or FPGA, each bit operation can be efficiently performed individually. Because the distribution of heterogeneous weight bits will be fixed at inference time (activations would be binarized homogeneously), fixed gates can be allocated depending on the bitwidth of individual values. This addresses the challenge of sparsity and allows a heterogeneous bitwidth FPGA implementation to have fewer total gates and a lower power consumption than a fixed bitwidth implementation.

## 4 EXPERIMENTS

To evaluate HBNNs we wished to answer the following four questions:

(1) How does accuracy scale with an uninformed bit distribution?

(2) How do the bit selection methods of Equation 6 compare?

(3) How well do HBNNs perform on a challenging dataset compared to the state of the art?

(4) Can the benefits of HBNNs be transfered to other architectures?

In this section we address each of these questions.

### 4.1 IMPLEMENTATION DETAILS

AlexNet with batch-normalization (AlexNet-BN) is the standard model used in binarization work due to its longevity and the general acceptance that improvements made to accuracy transfer well to more modern architectures. Batch normalization layers are applied to the output of each convolution block, but the model is otherwise identical to the original AlexNet model proposed by Krizhevsky et al. (2012). Besides it's benefits in improving convergence, batch-normalization is especially important for binary networks because of the need to equally distribute values around zero. We additionally insert binarization functions within the convolutional layers of the network when binarizing weights and at the input of convolutional layers when binarizing inputs. We keep a floating point copy of the weights that is updated during back-propagation, and binarized during forward propagation as is standard for binary network training. We use the straight-through estimator for gradients.

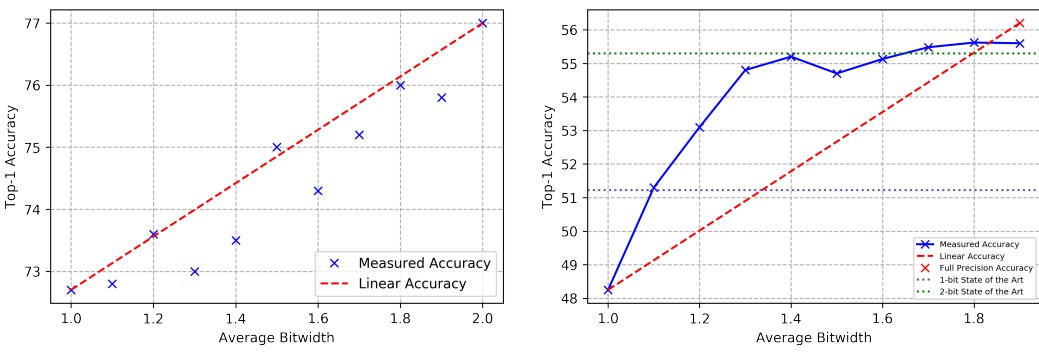

(a) CIFAR-10 uninformed bit selection.   (b) HBNN AlexNet with Middle-Out bit selection.

Figure 3: Accuracy results of trained HBNN models. (a) Sweep of heterogenous bitwidths on a deliberately simplified four layer convolutional model for CIFAR-10. Bits are distributed randomly at a kernel level. Trained accuracy tracks very linearly with bitwidth. (b) Accuracy of heterogeneous bitwidth AlexNet-BN models. Bits are distributed using the Middle-Out selection algorithm. Here, the performance is much better than linear, implying Middle-Out selection effectively chooses which values need more bits and which do not.

When binarizing the weights of the network's output layer, we add a single parameter scaling layer that helps reduce the numerically large outputs of a binary layer to a size more amenable to softmax, as suggested by Tang et al. (2017). We train all models using an SGD solver with learning rate 0.01, momentum 0.9, and weight decay 1e-4 and randomly initialized weights for 90 epochs, using the PyTorch framework.

## 4.2 MICROBENCHMARKS

Here we conduct two simple experiments to measure the ability of various binarization schemes to approximate a floating point tensor.

### 4.2.1 KERNEL-LEVEL HETEROGENEITY

As a baseline, we test a "poor man's" approach to HBNNs, where we fix up front the number of bits each kernel is allowed, require all values in a kernel to have its associated bitwidth, and then train as with conventional, homogeneous binarization. We consider 10 mixes of 1, 2 and 3-bit kernels so as to sweep average bitwidths between 1 and 2. We trained as described in Section 4.1. For this experiment, we used the CIFAR-10 dataset with a deliberately hobbled (4-layer fully conventional) model with a maximum accuracy of roughly 78% as the baseline 32-bit variant. We chose CIFAR-10 to allow quick experimentation. We chose not to use a large model for CIFAR-10, because for large models it is known that even 1-bit models have 32-bit-level accuracy Courbariaux et al. (2016).

Figure 3a shows the results. Essentially, accuracy increases roughly linearly with average bitwidth. Although such linear scaling of accuracy with bitwidth is itself potentially useful (since it allows finer grain tuning on FPGAs), we are hoping for even better scaling with the "data-aware" bitwidth selection techniques of Equation 5.

### 4.2.2 CHARACTERIZING REPRESENTATIONAL POWER

To compare the approximating capability of the selection methods of Equation 6, we generate a large tensor of normally distributed random values and apply top-down, bottom-up, middle-out, and random binarization with a variety of bit mixes to binarize it. We compare the normalized Euclidean distance of the approximated tensors with the input as a measure of how well the input is approximated. We additionally binarize the tensor using homogeneous bitwidth approximation to gauge the representational power of previous works.

The results, shown in Figure 2a, show that middle-out selection vastly outperforms other methods. In fact, it is clear that 1.4 bits distributed with the middle-out algorithm approximates the input roughly

Table 1: Accuracy of related binarization work and our results using AlexNet architecture

|  | Model | Binarization (Inputs / Weights) | Top-1 | Top-5 |
|---|---|---|---|---|
|  | Binarized weights with floating point activations | | | |
| 1 | SQ-BWN (Dong et al., 2017) | full precision / 1 bit | 51.2% | 75.1% |
| 2 | SQ-TWN (Dong et al., 2017) | full precision / 2 bit | 55.3% | 78.6% |
| 3 | TWN (our implementation) | full precision / 1 bit | 48.3% | 71.4% |
| 4 | TWN | full precision / 2 bit | 54.2% | 77.9% |
| 5 | HBNN (our results) | full precision / 1.4 bit | 55.2% | 78.4% |
|  | Binarized weights and activations excluding first and last layers | | | |
| 6 | BNN (Courbariaux et al., 2015) | 1 bit / 1-bit | 27.9% | 50.4% |
| 7 | Xnor-Net (Rastegari et al., 2016) | 1 bit / 1 bit | 44.2% | 69.2% |
| 8 | DoReFaNet (Zhou et al., 2016) | 2 bit / 1 bit | 50.7% | 72.6% |
| 9 | QNN (Hubara et al., 2016) | 2 bit / 1 bit | 51.0% | 73.7% |
| 10 | AlexNet (our implementation) | 2 bit / 2 bit | 52.2% | 74.5% |
| 11 | AlexNet | 3 bit / 3 bit | 54.2% | 78.1% |
| 12 | HBNN | 1.4 bit / 1.4 bit | 53.2% | 77.1% |
| 13 | HBNN | 1 bit / 1.4 bit | 49.4% | 72.1% |
| 14 | HBNN | 1.4 bit / 1 bit | 51.5% | 74.2% |
| 15 | HBNN | 2 bit / 1.4 bit | 52.0% | 74.5% |
|  | Unbinarized (our implementation) | | | |
| 16 | Alexnet (Krizhevsky et al., 2012) | full precision / full precision | 56.5% | 80.1% |

as well as standard 2-bit binarization. Also interesting to note is that the actual bit mix changes performance rather than just the average. A mix of 40%/50%/10% 1-bit/2-bit/3-bit provides quite a bit better than a mix of 50%/30%/20% 1-bit/2-bit/3-bit despite both having an average of 1.7 bits. This difference suggests that some bit mixtures may be better suited to approximating a distribution than others.

To measure the transfer of the representational benefits of Middle-Out selection to a real network training scenario, we apply each of the selection techniques discussed to the AlexNet-BN model described in Section 4.3 and train with the same hyper-parameters as in Section 4.1. We binarize the weights of each model to an average of 1.4 bits using 70% 1 bit values, 20% 2 bit values, and 10% 3 bit values. For random distribution, we sample from a uniform bitwidth distribution before training and then fix those values. Unlike in CIFAR-10, bits are randomly distributed *within* kernels as well.

The results are shown in Figure 2b. Quite clearly, Middle-Out selection outperforms other selection techniques by a wide margin, and is in fact roughly the same accuracy as using a full two bits. Interestingly, the accuracy achieved with Bottom-Up selection falls on the linear projection between 1 and 2 bits. Random and Top-Down distribution perform below the linear. Thus, Middle-Out selection seems to be the only technique that allows us to achieve a favorable trade-off between accuracy and bitwidth and for this reason is the technique we focus on in the rest of our experiments.

### 4.3 ALEXNET WITH BINARIZED WEIGHTS AND NON-BINARIZED ACTIVATIONS

Recently, Dong et al. (2017) were able to binarize the weights of an AlexNet-BN model to 2 bits and achieve nearly full precision accuracy (row 2 of Table 1). We consider this to be the state of the art in weight binarization since the model achieves excellent accuracy despite all layer weights being binarized, including the first and last layers which have traditionally been difficult to approximate. We perform a sweep of AlexNet-BN models binarized with fractional bitwidths using middle-out selection with the goal of achieving comparable accuracy using fewer than two bits.

The results of this sweep are shown in Figure 3b. We were able to achieve nearly identical top-1 accuracy to the best full 2 bit results (55.3%) with an average of only 1.4 bits (55.2%). As we had hoped, we also found that the accuracy scales in a *super-linear* manner with respect to bitwidth when using middle-out compression. Specifically, the model accuracy increases extremely quickly from

1 bit to 1.3 bits before slowly approaching the full precision accuracy. We explored many different mixes of bits that gave the same average bitwidth, but found that they gave nearly identical results, suggesting that when training from scratch the composition of a bit mix does not matter nearly so much as the average number of bits. Our 1-bit performance is notably worse, perhaps because we did not incorporate the improvements to training binary nets suggested by Dong et al. (2017). Adding stochastic layer binarization may have boosted our low-bitwidth results and allowed us to achieve near full precision accuracy with an even lower bitwidth.

To confirm that heterogeneous binarization can transfer to state of the art networks, we apply 1.4 bit binarization with 70% 1 bit, 20% 2 bit, and 10% 3 bit values to MobileNet (Howard et al., 2017), a state of the art architecture that achieves 68.8% top-1 accuracy. To do this, we binarize the weights of all the depthwise convolutional layers (i.e., 13 of 14 convolutional layers) of the architecture to 1.4 bits using middle-out selection and train with the same hyper-parameters as AlexNet. Our HBNN reached a top-1 accuracy of 65.1%.

### 4.4 ALEXNET WITH BINARIZED WEIGHTS AND ACTIVATIONS

In order to realize the speed-up benefits of binarization (on CPU or FPGA) in practice, it is necessary to binarize both inputs the weights, which allows floating point multiplies to be replaced with packed bitwise logical operations. The number of operations in a binary network is reduced by a factor of $\frac{64}{mn}$ where $m$ is the number of bits used to binarize inputs and $n$ is the number of bits to binarize weights. Thus, there is significant motivation to keep the bitwidth of both inputs and weights as low as possible without losing too much accuracy. When binarizing inputs, the first and last layers are typically not binarized as the effects on the accuracy are much larger than other layers. We perform another sweep on AlexNet-BN with all layers but the first and last fully binarized and compare the accuracy of HBNNs to several recent results. Row 7 of Table 1 is the top previously reported accuracy (44.2%) for single bit input and weight binarization, while row 9 (51%) is the top accuracy for 2-bit inputs and 1-bit weights.

Table 1 (rows 12 to 15) reports a selection of results from this search. Using 1.4 bits to binarize inputs and weights ($mn = 1.4 \times 1.4 = 1.96$) gives a very high accuracy (53.2% top-1) while having the same number of total operations $mn$ as a network, such as the one from row 7, binarized with 2 bit activations and 1 bit weights. We have similarly good results when leaving the input binarization bitwidth an integer. Using 1 bit inputs and 1.4 bit weights, we reach 49.4% top-1 accuracy which is a large improvement over Rastegari et al. (2016) at a small cost. We found that using more than 1.4 average bits had very little impact on the overall accuracy. Binarizing inputs to 1.4 bits and weights to 1 bit (row 14) similarly outperforms Hubara et al. (2016) (row 7, mentioned above); however, the accuracy improvement margin is smaller.

## 5 CONCLUSION

In this paper, we present Heterogeneous Bitwidth Neural Networks (HBNNs), a new type of binary network that is not restricted to integer bitwidths. Allowing effectively fractional bitwidths in networks gives a vastly improved ability to tune the trade-offs between accuracy, compression, and speed that come with binarization. We show a simple method of distributing bits across a tensor lead to a linear relationship between accuracy and number of bits, but using a more informed method allows higher accuracy with fewer bits. We introduce middle-out bit selection as the top performing technique for determining where to place bits in a heterogeneous bitwidth tensor and find that Middle-Out enables a heterogeneous representation to be more powerful than a homogeneous one. On the ImageNet dataset with AlexNet and MobileNet models, we perform extensive experiments to validate the effectiveness of HBNNs compared to the state of the art and full precision accuracy. The results of these experiments are highly compelling, with HBNNs matching or outperforming competing binarization techniques while using fewer average bits. The use of HBNNs enables applications which require higher compression and speeds offered by a low bitwidth but also need the accuracy of a high bitwidth. As future work, we will investigate modifying the bit selection method to make heterogeneous bit tensors more amenable for CPU computation as well as develop a HBNN FPGA implementation which can showcase both the speed and accuracy benefits of heterogeneous binarization.

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
