# OpenReview forum: "Heterogeneous Bitwidth Binarization in Convolutional Neural Networks"
_ICLR.cc/2018/Conference — Reject_

### Official Review · AnonReviewer1 · 2017-11-24
**There are flaws in comparisons with previous works**

**Rating:** 6
**Confidence:** 4

**Review:**

This paper suggests a method for varying the degree of quantization in a neural network during the forward propagation phase.

Though this is an important direction to investigate, there are several issues:

1. Comparison with previous results is misleading:
a.	1-bit weights and floating point activations: Rastegari et al. got 56.8% accuracy on Alexnet, which is better than this paper 1.4bit result of 55.2%.
b.	Hubara et al. got 51% results on 1-bit weights and 2-bit activations included also quantization first and last layer, in contrast to this paper. Therefore, it is not clear if there is a significant benefit in the proposed method which achieves 51.5% when decreasing the activation precision to 1.4bit.

Therefore, it is not clear that the proposed methods improve over previous approaches.

2. It is not clear to me: in which dimension of the tensors are we saving the scale factor? If it is per feature map, or neuron, this eliminates the main benefits of quantization: doing efficient binarized operations when doing Weight*activation during the forward pass?

3. The review of the literature is inaccurate. For example, it is not true that Courbariaux et al. (2016) “further improved accuracy on small datasets”: the main novelty there was binarizing the activations (which typically decreased the accuracy). Also, it is not clear if the scale factors introduced by XNOR-Net indeed allowed "a significant improvement over previous work" in ImageNet (e.g., see DoReFA and Hubara et al. who got similar results using binarized weigths and activations on ImageNet without scale factors).  Lastly, the statement “Typical approaches include linearly placing the quantization points” is inaccurate: it was observed that logarithmic quantization works better in various cases. For example, see Miyashita, Lee and Murmann 2016, and Hubara et al.

%%% After Author's Clarification %%%
This paper results seem more positive now, and I have therefore have increased my score, assuming the authors will revise the paper accordingly.

---

> ### Author Response · Authors · 2017-12-04
> **Reviewer 1 Response**
>
> Reviewer 1 points out flaws in comparisons with related work.
>
> [Rastegari shows 56.8% accuracy, we only show 55.2% accuracy (row 4 of Table 1)] We measure the result of binarizing *all* layers of Alexnet (similar to Dong et al, from rows 1 and 2). Rastegari et al *do not binarize the first or last layer*. We consider Dong's result to be more challenging and chose to compare to it. However, we are happy to also compare to Rastegari's configuration if reviewers think it is misleading not to. Or we can just make this difference (currently noted in section 4.3) explicit in the table.
>
> [Hubara got 51% with 1-bit weights/2-bit activations when binarizing all layers, whereas we got 51.5% without binarizing first and last layer] We admit that this partly slipped by us. We chose our binarization configuration to compare to the many other pieces of work in Table 1, none of which (to our understanding) binarize first and last layers. Please note however, that Hubara uses 8-bit binarization for the first layer, which is arguably closer to "no binarization" than to the conventional 1- and 2-bit binarization. Further, other work (e.g. Tang et al AAAI 2017) has shown that binarizing the last layer, unlike the first, does not result in much accuracy loss. But we are happy to report Hubara's configuration if reviewers deem this as misleading.
>
> [Are scale factors stored per feature map or neuron?] No, they are stored per kernel exactly as in Rastegari et al. Scale factor multiplication can be done after the binary product by multiplying each output feature map by it's corresponding scalar. The amount of added work is equivalent to replacing ReLU activations with PReLU activations, which does not have a significant effect on network inference time. We can make this point more explicit in the Implementation section.
>
> [Misunderstandings of key innovations and contributions in related work] We are embarrassed at our misunderstanding of the literature. We thank the reviewer and will correct these and improve our understanding.

---

> > ### Comment · AnonReviewer1 · 2017-12-31
> > **Correction**
> >
> > The statement "Hubara uses 8-bit binarization for the first layer" is not true. All layers in Hubara et al. were quantized to 1-bit weights and 2-bit activations, including the first layers (i.e. the first weight layer, and the first hidden layer). Perhaps the authors misunderstood: the input layer is indeed 8-bit, as always.
> >
> > Thanks for the other clarifications. Still, taking into consideration all the results together, it is not clear to me that they are significantly better then previous works.

---

> > > ### Author Response · Authors · 2018-01-01
> > > **First Layer Vs Input Layer**
> > >
> > > We seem to be having a little bit of a terminology mismatch. In our work, all references to first layer mean the input layer, just as references to the last layer mean the output layer. We see now how this can cause confusion and will update our lingo to exclusively use input layer and output layer.
> > >
> > > With this terminology mismatch in mind, our description of Hubara's work matches the implementation in his github repo (https://github.com/itayhubara/BinaryNet). The architectures in our work are comparable.
> > >
> > > Hopefully clarifying this point allows more direct comparison to other works and highlights the significance of our results.

---

### Official Review · AnonReviewer2 · 2017-11-27
**The improvement is not significant**

**Rating:** 5
**Confidence:** 4

**Review:**

The paper tries to maintain the accuracy of 2bits network, while uses possibly less than 2bits weights.

1.  The paper misses some more recent reference, e.g. [a,b]. The author should also have a discussion on them.

2. Indeed, AlexNet is a good seedbed to test binary methods. However, it is more interesting and important to test on more advanced networks. So, I wish to see a section on testing with Resnet and GoogleNet.

Indeed, the authors have commented: "AlexNet with batch-normalization (AlexNet-BN) is the standard model ... acceptance that improvements made to accuracy transfer well to more modern architectures." So, please show that.

3. The paper wants to find a good trade-off on speed and accuracy. The authors have plotted such trade-off on space v.s. accuracy in Figure 3(b), then how about speed v.s. accuracy?

My concern is that one-bit system is already complicated to implement. Indeed, the authors have discussed their implementation in Section 3.3, so, how their method works in practice? One example is Section 4 in [Courbariaux et al. 2016].

4. Is trade-off between 1 to 2 bits really important?

Compared with 2bits or ternary network, the proposed method at most achieving (1.4/2) compression ratio and (2/1.4) speedup (based on their Table 1). Is such improvement really important?

Reference:
[a]. Trained Ternary Quantization. ICLR 2017
[b]. Extremely low bit neural network: Squeeze the last bit out with ADMM. arvix 2017

---

> ### Author Response · Authors · 2017-12-04
> **Reviewer 2 Response**
>
>  The reviewer questions whether the performance improvements we claim are significant. In detail:
>
> [Comparison to related work] We thank the reviewer for these references. Reference A, on ternary networks, is quite similar to the work of Li et al that we reference in related work, but we will include further discussion. Reference B suggests training improvements (not binarization techniques), which we will look to adopt.
>
> [Evaluation in larger models: please show results on Resnet/GoogleNet] We selected AlexNet to illustrate most of the work, since it has been used exclusively in the community to compare approaches. However, do note that **unlike any paper so far**, we have shown results also on Mobilenet, which is the state-of-the art object recognition model as of fall 2017, from Google (contribution 4 in the intro, and last paragraph of section 4.3). Mobilenet yields comparable accuracy to Resnet and GoogleNet, but is also much faster than them, and is therefore a challenging benchmark. Perhaps we should highlight this result better?
>
> [Complexity of implementation] Gaining performance from binarized models is indeed complex, especially on a CPU. In fact, no paper provides the many crucial details, such as machine specific vectorization/tiling/loop fusion algorithms essential to gaining real-world speedup. Even Courbariaux et al, section 4, only gives a sketch in this direction. Admittedly, heterogeneous bitwidths will add to this complexity, so this is a fair concern on a CPU. However, implementation is fairly straightforward on an FPGA, because we simply lay out custom gate patterns for each bit pattern to be XNOR'd against (see e.g. https://arxiv.org/pdf/1612.07119.pdf , esp. section 4.3.2 for a similar implementation in the homogeneous bitwidth case). The custom pattern for processing 2 bits is only slightly different from the 1-bit version. Perhaps we can focus the implementation section on sketch how to perturb this standard FPGA-based design?
>
> [No speed vs accuracy number] As mentioned above, almost no paper in this area reports measured speedups, just improvements in coarsely estimated instruction counts. In the FPGA context, we could similarly report coarse estimates the number of cycles, chip real estate and power consumed. However, roughly speaking these (especially the latter two that are our goal) are simply proportional to average bitwidth of the operations programmable into hardware. We could make this explicit in the text when we discuss the implementation above.
>
> [A 1.4/2x = 0.7x reduction is not significant] Although this is a subjective call and hard to argue against, it is worth noting that our gains are *on top of* optimized binary implementations. Further, note that FPGA implementations of DNNs are now running at cloud scale e.g. in the Azure cloud (https://www.microsoft.com/en-us/research/blog/microsoft-unveils-project-brainwave/). A 30% improvement in space/energy efficiency with no accuracy loss is considered quite significant at these scales.

---

### Official Review · AnonReviewer3 · 2017-12-01
**Review of Heterogeneous Bitwidth Binarization in Convolutional Neural Networks**

**Rating:** 4
**Confidence:** 4

**Review:**

This paper presents an extension of binary networks, and the main idea is to use different bit rates for different layers so we can further reduce bitrate of the overall net, and achieve better performance (speed / memory). The paper addresses a real problem which is meaningful, and provides interesting insights, but it is more of an extension.

The description of the Heterogeneous Bitwidth Binarization algorithm is interesting and simple, and potentially can be practical, However it also adds more complication to real world implementations, and might not be an elegant enough approach for practical usages.

Experiments wise, the paper has done solid experiments comparing with existing approaches and showed the gain. Results are promising.

Overall, I am leaning towards a rejection mostly due to limited novelty.

---

> ### Author Response · Authors · 2017-12-04
> **Reviewer 3 Response**
>
> The reviewer describes our work as seeking to use different bit rates for different layers, and points out that the work is not novel enough overall.
>
> We would like to point out that in fact, we are not looking to simply binarize different layers at different bitwidths (although as a baseline we do so in figure 3(a)). In that baseline, assuming we select *up front* what the bitwidth k of each layer is, we use the not-so-novel approach of simply applying standard k-bit binarization algorithms with different k to each layer. This is indeed simple to do, but figure 3(a) shows that such naive selection only provides "linear" increase in accuracy (e.g., using 1.5 bits on average gives only the average of 1-bit and 2-bit accuracies, which is interesting but perhaps not surprising).
>
> Instead, in our main contribution, we are asking the question "if we *learned* what bitwidth to assign to *each* parameter (jointly with its value), could we get better-than-linear speedup". This learning of  bitwidths  is what is novel about our goal, and techniques.
>
> Learning bitwidths requires changing the training algorithm in a non-obvious way, using the mask-generation scheme of algorithm 1, and the middle-out scheme for thresholding. We should emphasize that the operations of equation 4, Algorithm 1 and equation 6 are not performed in a one-time "post-processing" step, but on every forward propagation during training. We submit that this learning algorithm is quite novel.
>
> Given that the bitwidth is such a fundamental aspect of model parameters, we hope that learning them jointly with values should be of broad interest to the ICLR community.

---

### Public Comment · ~Angus_Galloway1 · 2017-12-02
**Comparison with BitNet would be helpful**

A comparison with BitNet -- (https://arxiv.org/pdf/1708.04788.pdf) would be helpful. Although they do not go down to the very low precision (1-2 bit) case as with your paper, they do learn a unique precision for the parameters of each layer via SGD.

---

> ### Author Response · Authors · 2017-12-04
> **BitNet Comparison**
>
> Thanks for pointing us to this work. We weren’t aware of it, and it looks interesting and related. More details below.
>
> Our two papers focus on different problems, but have quite a bit of potential to improve each other in future work. BitNet presents a method for learning the integer number of bits to binarize each layer in a network while our work presents a method for learning the number of bits for each individual parameter within a layer. In other words, BitNet is learning bitwidths with layer granularity while we're learning bitwidths with parameter level granularity.
>
> To dig into this difference a little deeper, noting figure 1 in BitNet, we can see that during training the bitwidth of each layer is quite noisy due to the large discontinuity between bits. This is because prior to our work, there was no method to binarize a tensor to a fractional bitwidth. If the authors of BitNet were to incorporate Heterogeneous Bitwidth layers, the figure 1 of BitNet could be made smooth and continuous. This should help quite a bit with improving the stability of BitNet's learning process.
>
> Beyond the fundamental contribution of each paper, our work also differs by binarizing to lower bitwidths. The majority of layers in BitNet end up being binarized to around 6 bits, which is a huge difference in representational power than the 2 bit and below widths we focus on. Traditionally, showing good performance on 1-2 bits has been significantly harder than e.g. 4 bits and above.
>
> Additionally, BitNet only focuses on the CIFAR and MNIST datasets. Many works have found that success on these datasets does not always transfer to more difficult datasets such as ImageNet. In contrast we provide a detailed analysis of results of ImageNet classification, including for the first time, a state-of-the-art model, Google’s Mobilenet.

---

### Author Response · Authors · 2017-12-04
**Rebuttal Summary**

 We thank the reviewers for their detailed and useful feedback. Below, we make two summary points in response and follow up with some detailed responses to the issues raised.

At the highest level,  we realize from the feedback we did not make the motivation and the technical significance of our work sufficiently clear.

First, the motivation. We are looking to adapt binarization algorithms for implementation on FPGAs. On standard FPGA-based implementations (e.g., https://arxiv.org/pdf/1612.07119.pdf ), every XNOR product is implemented as a separate hardware structure proportional to the number of bits in the product. Typically, both the real estate (number of gates) and power consumed (watts) by an implementation are proportional to the average bitwidth of the data. A (2 to 1.4 =) 30% average reduction in real estate and power **at no accuracy loss** is quite attractive. We intend to rework our introduction and implementation sections to highlight this perspective.

Second, the significance. Traditionally, ML algorithms look to optimize/learn the *value* of every parameter. We extend the optimization criterion in a simple, but fundamental, way to include the representation. We ask whether it is (a) feasible and (b) useful to *jointly* learn both the value *and the bitwdith* of every parameter.  We provide an affirmative answer on both counts, and to our knowledge, are the first to do so. This advance requires a non-obvious change to the training scheme (i.e., the middle-out scheme to select a variable-bitwidth mask, Algorithm 1). We also show experimentally (sec 4.2.1 and Figure 3a) that a simpler approach that does not pick the bitwidths in a data driven manner does not give the same bump in performance. Again, we will reframe our paper to highlight this.

Detailed responses are provided below to each reviewer.

---

### Decision · Program_Chairs · 2018-01-29
**ICLR 2018 Conference Acceptance Decision**

**Decision:**

Reject

**Comment:**

All of the reviewers find the approach interesting, but they have reservations regarding the practical impact and empirical evaluation. The paper needs improvement both on the motivation and on the experimental results by including more baseline methods and neural architectures.